# On the Road to Fight Cancer: The Potential of G-Quadruplex Ligands as Novel Therapeutic Agents

**DOI:** 10.3390/ijms22115947

**Published:** 2021-05-31

**Authors:** Irene Alessandrini, Marta Recagni, Nadia Zaffaroni, Marco Folini

**Affiliations:** Molecular Pharmacology Unit, Department of Applied Research and Technological Development, Fondazione IRCCS Istituto Nazionale dei Tumori di Milano, Via G.A. Amadeo 42, 20133 Milan, Italy; irene.alessandrini2@istitutotumori.mi.it (I.A.); marta.recagni@istitutotumori.mi.it (M.R.); nadia.zaffaroni@istitutotumori.mi.it (N.Z.)

**Keywords:** anticancer therapy, cell defense mechanisms, cancer, gene promoters, G-quadruplex, synthetic lethality, telomeres

## Abstract

Nucleic acid sequences able to adopt a G-quadruplex conformation are overrepresented within the human genome. This evidence strongly suggests that these genomic regions have been evolutionary selected to play a pivotal role in several aspects of cell biology. In the present review article, we provide an overview on the biological impact of targeting G-quadruplexes in cancer. A variety of small molecules showing good G-quadruplex stabilizing properties has been reported to exert an antitumor activity in several preclinical models of human cancers. Moreover, promiscuous binders and multiple targeting G-quadruplex ligands, cancer cell defense responses and synthetic lethal interactions of G-quadruplex targeting have been also highlighted. Overall, evidence gathered thus far indicates that targeting G-quadruplex may represent an innovative and fascinating therapeutic approach for cancer. The continued methodological improvements, the development of specific tools and a careful consideration of the experimental settings in living systems will be useful to deepen our knowledge of G-quadruplex biology in cancer, to better define their role as therapeutic targets and to help design and develop novel and reliable G-quadruplex-based anticancer strategies.

## 1. Introduction

At the beginning of the twentieth century, it was suggested that guanine-rich DNA sequences have the potential to form high-order structures [1]. X-ray diffraction studies then revealed that guanylic acids may assemble into tetrameric structures that are responsible for the gel-like properties of polymeric runs of guanylic acid in aqueous solution [1]. Subsequently, it was reported that four molecules of guanylic acid may form a square planar arrangement (G-quartet) in which the guanine residues are hydrogen-bonded to each other via Hoogsteen pairings, in the presence of monovalent cations (Figure 1) [1]. The stacking of two or more G-quartets on top of each other generates a peculiar non-B DNA conformation, referred as to G-quadruplex structure (G4) (Figure 1) [1,2]. G4 structures may form within G-rich nucleic acid sequences under physiological conditions [1,2,3]. They are highly thermodynamically stable and exhibit extensive structural polymorphism [4,5]. They may indeed (i) generate from one nucleic acid strand (intramolecular G4) as well as from two (bimolecular) or four (tetramolecular) separate strands, thus generating intermolecular G4s (Figure 1); (ii) be classified, depending on the orientation of the nucleic acid strands connecting the guanines, in parallel, antiparallel or hybrid structures; (iii) be characterized by the nature and length of G-G connecting loops, the syn vs. anti glycosil conformation as well as the number of stacking quartets [1,2,3,4,5].

A large number of human genomic G-rich sequences able to fold into G4 have been described thus far, including telomeric DNA, gene promoters, replication origins and 5’-untranslated regions [2,5]. Telomeres, which are located at the very ends of human chromosomes and are composed of exameric G-rich sequences (5’-TTAGGG-3’ in vertebrates) repeated in tandem, were the first biologically relevant G4 forming sequences to be studied in details [5,6]. In particular, it has been documented that telomeric G4 may cap telomeres, thus protecting them from inappropriate elongation by telomerase or from nucleolytic degradation and end-to-end fusion events [6]. Furthermore, telomeric G4 may act as a barrier for the recombination events that sustain the alternative lengthening of telomeres (ALT) pathway [6], a telomere maintenance mechanism that operates in certain type of human tumors lacking telomerase activity [7]. In addition, telomere-associated proteins, such as the heterodimer formed by the shelterin components POT1 and TPP1, have been found to bind and unwind telomeric G4 [8]. These observations together with the evidence that telomeric DNA can fold into different quadruplex structures (structural polymorphism) and that the different G4 topologies may mutually be in a dynamic equilibrium [9], support the notion that G4 structures may play an important biological role in the regulation of telomere function [6].

Sequences showing a consensus motif consistent, at least in theory, with the capability to generate G4 structures have been found to be conserved during evolution [10]. In particular, quadruplex-forming sequences appears to be over-represented throughout the human genome [2,5,11], to the point that the term *G4 genome* was proposed some years ago to indicate such a vast repertoire of G4 forming motifs [11]. This evidence alongside the observation that genomic regions spanning ~1000 nucleotides upstream the transcription start sites are chiefly enriched in quadruplex forming sequences [10,11] has strongly supported the idea that these regions might be evolutionary selected to play a functional role in the regulation of gene expression [11]. In addition, G4-forming sequences are frequently found in the promoters of oncogenes and transcription factors rather than in tumor suppressor or housekeeping genes thus indicating an evolutionary selection of these elements based on gene function [10,11]. Notably, ChIP-seq analyses carried out using an anti-G4 structure antibody has recently revealed the occurrence of ~10,000 G4 elements in regulatory, nucleosome-depleted chromatin regions (e.g., promoters and 5’-UTRs) of highly transcribed genes [12].

As originally outlined by Brooks et al. [13], it is now possible to catalog several tumor-associated genes, the promoter of which may harbor G4 forming sequences [13,14,15]. Consequently, in the perspective of antitumor therapeutic strategies, the selective stabilization of G4 within gene regulatory elements by small-molecules has appeared as a suitable tool to operate at genomic level for the control of aberrant gene expression [15,16,17,18]. This notion has gained support by data showing that the G4 ligand pyridostatin (PDS) was able to inhibit gene expression by interacting with quadruplex-forming sequences within the human genome [19]. Specifically, it was observed that PDS is able to target gene bodies containing clusters of quadruplex-forming sequences and that the drug-induced sites of DNA damage corresponded to target genes that were all down-regulated following the exposure to PDS [19]. Subsequent biophysical analyses have evidenced that the quadruplex-forming sequences within the SRC gene, which was identified as the most down-regulated gene upon treatment with PDS, were able to adopt a stable G4 conformation and that PDS selectively interacted with them through a stacking mode [19].

Finally, pieces of evidence have shown that G4 structures may be relevant also in the RNA world [20]. Indeed, RNA regions, such as the 5’-untranslated regions (5’-UTR) [11], or RNA species, including long non-coding RNAs (lncRNAs) [21], have been found to be enriched in quadruplex-forming sequences. This evidence indicates that RNA G4 may play an important regulatory role at post-transcriptional level, where they can operate in *cis* or in *trans* to control the coding capacity of the genome [20,21]. Specifically, G4 structures located in the proximity of splicing sites can act as *cis*-regulatory elements during the splicing reaction of a number of genes, including the tumor suppressor *TP53* and the human telomerase reverse transcriptase *TERT* [22].

Computational analyses have revealed that G4 forming motifs are over-represented in the 5′-UTR of mRNAs [20]. The presence of G4 structures at this location may affect the translation initiation thus resulting in the inhibition of protein production [22]. However, it has been also reported that G4 located in 5′-UTR may favor active translation, as is the case of the G4-forming sequences within the internal ribosome entry site (IRES) elements, such as those of the vascular endothelial growth factor (*VEGF*) and of fibroblast growth factor 2 (*FGF2*) mRNAs [22]. In addition, G4 forming sequences within the 3’-UTR, though occurring to a significantly low frequency with respect to those found in the 5’-UTR [11,20], has been regarded as *cis*-acting elements that may regulate the sub-cellular sorting of mRNAs [22].

Finally, RNA G4 have been found also in lncRNAs [23]. For instance, the G-rich element at the 5′ end of *TERC*—the RNA moiety of telomerase that contains the template sequence for the de novo synthesis of telomeric repeats—may fold into a G4. It has been assumed that this structure protects ***TERC*** from degradation during the early stages of telomerase ribonucleoprotein biogenesis and impedes the formation of a helix domain required for the definition of the template boundary in mammalian telomerase [22]. Moreover, owing to its G-rich nature, the lncRNA transcribed from telomeric DNA (TERRA) may form hybrid G4 structures with telomeric DNA. The formation of these intermolecular G4 at telomeric level sems to act as a physical barrier that interferes with the telomere extending activity of telomerase [22].

## 2. Targeting G-Quadruplex Structures in Cancer

Although G4 structures could appear as elusive entities and their biological role still requires to be elucidated in depth, the evidence that these structures (i) are stable and detectable in human genomic DNA [24]; (ii) can be detected in living cells by specific antibodies or fluorescent probes [25]; (iii) are prevalent in tumor than in normal tissues [26,27], alongside the evidence that some human genetic disorders are driven by defects in the unwinding activity of G4-associated enzymes [28], indicates that G4 may represent suitable druggable targets. Indeed, during the past two decades several efforts have been made in the search of small molecules able to recognize and bind to G4 structures [2,15,18,29]. These compounds broadly share defined structural features, including (1) a large aromatic core for π-π stacking interactions with the G-quartet; (2) one or more flexible side-chains on the aromatic core that allow introducing specific functional groups to favor the interaction with grooves, loops or individual bases of the G4 [30,31,32]. Moreover, the presence of one or more positive charges may enable electrostatic interaction with the anionic charges of the nucleic acid backbone, though this may result in a little ability to recognize specific G4 topologies [2]. The peculiar 3D structure of G4 allows the recognition by small molecules by various binding modes. In particular, G4 ligands may (1) stack externally on the surface of the terminal G-quartet; (2) stack on the side of the G4; (3) intercalate between the stacks of G-quartets and (4) bind to the grooves. A combination of two or more of these binding modes can give rise to various degrees of binding selectivity [30]. It has been also reported that compounds with reduced planarity (i.e., lacking the flat aromatic core) may be amenable of G4 targeting through the interaction with the grooves and/or the nucleic acid backbone [33]. To date, several small molecules belonging to distinct chemical families have been designed and characterized as G4 ligands [34] and a subset of them have been evaluated for their therapeutic potential in cancer [14,15,16,17,18,29].

### 2.1. Casting a Glance over Telomeric G4 Ligands

Telomeric DNA has been the first biologically relevant target to be tested for small-molecule-mediated G4 stabilization [35]. The search for telomeric G4 ligands has been fostered by the G-richness of telomeric DNA and its propensity to fold into G4 structures [36] as well as by the idea that drug-mediated stabilization of telomeric G4 would have allowed to selectively interfere with the telomere lengthening activity of human telomerase in cancer [35].

To date, a myriad of small molecules have been tested for their capability to recognize and stabilize telomeric G4, even regardless of their specificity for telomere over other genomic quadruplex forming sequences [14,17,32]. Among these compounds, the most advanced in terms of preclinical investigation has been the acridine derivative RHPS4 (3,11-difluoro-6,8,13-trimethyl-8H-quino[4,3,2-kl]acridinium methosulfate) [34]. The molecule was primarily documented to show a high selectivity for telomeric G4 structures and to exert good anticancer activity in several human tumor models [14]. Other than being active as single agent, RHPS4 has shown synergistic pharmacological interactions in vitro when combined to conventional anticancer agents [14] as well as to enhance the effects of carbon ion [37] and photon [38] radiotherapy. Furthermore, the compound showed a promising antitumor activity on human tumor models grown in mice both as single agent [14] and when combined with conventional anticancer drugs, such as taxanes [39] and camptothecins [40,41], as well as radiation [42]. Nonetheless, although RHPS4 showed a good therapeutic index, as it was well tolerated and did not cause general toxicity when administered to mice, its further clinical development has been halted by the evidence that the compound may induce undesirable side-effects on the cardiovascular system [43].

Literature data gathered thus far has allowed delineating a leading mechanism of action of telomeric G4 ligands in tumor models. Specifically, small-molecule-mediated stabilization of telomeric G4 has been repeatedly observed to produce two main and sometimes interconnected outcomes (Figure 2). In particular, long-term effects consequent to gradual telomere erosion, eventually leading to irreversible growth arrest (replicative senescence) of tumor cells, have been reported as the result of the indirect inhibition of telomerase activity in the presence of telomeric DNA locked into a G4 [14]. Moreover, short-term effects eventually leading to apoptotic cell death have been also described as a consequence of G4 ligand-mediated telomere uncapping, induction of telomeric DNA damage, impairment of fork progression and hence of telomere processing [14].

Despite a significant antitumor and chemo- and radio-sensitizing activity has been well documented in preclinical models for an ample number of telomeric G4 ligands, none of them have currently entered clinical trials. Nevertheless, the clinical significance of targeting telomeric G4 may gain support by the evidence that some of the clinically available chemotherapeutic drugs, or their derivatives, may possess G4-interacting capabilities.

The anthraquinone moiety is the scaffold of anthracyclines (e.g., doxorubicin; daunorubicin, and epirubicin), a class of antitumor antibiotics that interact with DNA and are widely used in different chemotherapeutic regimens for the clinical management of various types of cancers [44]. Notably, Compound 1 (2,6-diamidoanthraquinone derivative) was the first ligand reported to show selectivity towards telomeric G4 structures and to inhibit telomerase activity [35]. Since then, several reports have shown that anthracyclines, such as doxorubicin, daunomycin and epirubicin, can interact with telomeric DNA and may bind and stabilize telomeric G4 [45,46,47,48,49,50].

Cis-diamminedichloroplatinum (II) (Cisplatin) is a classical chemotherapeutic drug used to treat a number of cancers, including testicular, ovarian and breast cancers, mesothelioma, brain tumors and neuroblastoma. The drug roughly belongs to the family of alkylating agents [44], the general mode of action of which is the formation of an highly unstable ion intermediate which ultimately forms covalent bonds with DNA bases, the most vulnerable to attack being guanine [44]. Under physiological conditions, cisplatin molecules form positively charged active diaquated intermediates that attack the N7 atoms of purines to form mono- or di-adducts. Owing to their G-rich nature, G4 were proposed as promising targets for cisplatin as the N7 atom of the guanines out of the stack of the G-quartets should react with electrophilic species [51]. However, different biophysical methods used to investigate the interaction between cisplatin and telomeric G4 have shown that cisplatin may destabilize human telomeric G4 [52,53] and that cisplatin-mediated DNA platination, though occurring at significantly less frequency in telomeric vs. genomic DNA [54], markedly affected telomeric G4 folding [51]. Nonetheless, Ju et al., have reported that cisplatin and G4 may interact in two different and competitive ways [55]. Specifically, they demonstrated that cisplatin can bind reversibly to G4 structures and irreversibly to guanine residues present in the quadruplex-forming sequences, hence avoiding G4 folding [55]. These two types of interactions compete with each other depending on cisplatin concentration, with the reversible binding dominating over the irreversible one at low drug concentrations [55].

The elective affinity of cisplatin for guanine residues has fostered the search for platinum derivatives with improved G4 selectivity. In this regards, Tetra-Pt(bpy), which is composed of self-assembled Pt(II) molecular squares linked with 4-4’-dipyridyl bridges, was identified as a telomeric G4 ligand through the screening of a library of cisplatin derivatives [56]. By biophysical assays it was shown that the compound induced the formation of a parallel telomeric G4 under near-physiological conditions and that the melting temperature of this G4 structure was about 25 °C higher in the presence of the drug than its absence, thus indicating that the telomeric G4 was efficiently stabilized by the compound [56]. The exposure of ALT-positive osteosarcoma cells to Tetra-Pt(bpy) inhibited the strand invasion/annealing step of telomeric homologous recombination (HR) thus resulting in reduced telomere sister chromatin exchanges as well as in a decrease in the number of telomere-localized RPA and RAD51 foci, which indicate a reduction in HR intermediates. Moreover, a significant decrease in ALT-associated markers (i.e., C-circle DNA and ALT-associated Promyelocytic Leukemia Bodies (APBs)) were observed in Tetra-Pt(bpy)-treated ostesarcoma cells growing in vitro, indicating a drug-mediated suppression of ALT activity [56]. As a consequence, critically short telomeres accumulated after multiple population doublings in treated vs. untreated cells, resulting in apoptotic cell death or senescence. Importantly, Tetra-Pt(bpy) did not impair the proliferation and survival of normal human fibroblasts. Furthermore, Tetra-Pt(bpy) remarkably inhibited the growth of ALT-positive ostesarcoma cell grown as xenografts in mice and reduced the number of liver metastasis without causing general toxicity [56].

### 2.2. Impairment of G4-Mediated Regulation of Gene Expression for Therapeutic Purposes: The Paradigm of MYC

To date, the transcriptional activity of an ample number of cancer-associated genes has been reported to be amenable of G4-mediated regulation [2,14,15,17]. The finding that the expression of the proto-oncogene *MYC* (V-myc avian myelocytomatosis viral oncogene homolog) may undergo a G4-mediated regulation has been paradigmatic and has furnished the proof-of-concept for the G4-dependent regulation of gene expression [57,58].

Briefly, *MYC* is a transcription factor that regulates the expression of a variety of genes and is one of the most prevalent oncogene found to be altered in human cancer [59]. Aberrant MYC expression in cancer is mainly regulated at transcriptional level [57] by a complex mechanism involving four promoters (P1–P4), different transcription start sites and nuclease hypersensitive elements (NHE) [59]. In particular, the NHE III_1_, located just upstream the promoter P1 is responsible for 80-90% of *MYC* transcriptional activity [59]. This element is composed of five consecutive runs of a G-rich sequence that may fold into a G4 structure [57]. In this frame, it has been demonstrated that *MYC* transcription may be repressed following the stabilization of the G4 that may form within the gene promoter and that a single point mutation (G→T), which destabilizes the folding of such a G4, results in a three-fold increase in *MYC* basal transcriptional activity [57].

The elucidation of *MYC* G4 structure has led to the discovery of a great variety of molecules able to bind and stabilize it, both directly and indirectly, as elegantly reviewed in [2,57]. Recently, the bisacridine derivative a9 has been reported to directly bind and stabilize *MYC* G4, resulting in the downregulation of *MYC* transcription [60]. The exposure of squamous cell carcinoma cells to a9 resulted in the inhibition of cell growth paralleled by cell cycle arrest and apoptosis induction. Notably, the compound exhibited an antitumor activity on a xenograft model of squamous cell carcinoma that could be related to its binding to *MYC* G4, [60]. Similarly, the benzofuran derivative D089 has been reported to inhibit *MYC* transcription and to interfere with the survival of multiple myeloma cells through the direct binding to *MYC* G4 [61]. In particular, the exposure of myeloma cells to D089 induced endoplasmic reticulum stress, thus resulting in pyroptotic cell death, as evidenced by the appearance of a “ballooning” morphology associated with increased levels of the cleaved form of the inflammatory caspase 1 and of IL1-β as well as of gasdermin D cleavage and of HMGB1 cytoplasm translocation [61]. Recently, a Curcumin derivative, the synthetic 3,4-dimethoxy benzaldehyde Cur-4, has been recognized as a promising candidate for direct G4-mediated inhibition of *MYC* expression [62]. In particular, Cur-4 exhibited strong affinity and selectivity for G4 over duplex *MYC* DNA by binding at terminal G-tetrads [62]. The compound was found to significantly reduce the expression of *MYC* and to exert an anti-cancer activity in a panel of cancer cells as well as in multicellular tumor spheroid models [62].

It is now widely recognized that the regulation of *MYC* transcription is based on a fine interplay between transcription factors and the dynamic negative superhelicity induced during transcription [57]. The elucidation of how such a transcriptional machinery works has provided the first-in-class example of a novel level of complexity in gene transcription as well as the very first evidence of the existence of G4-protein interactions in living cells [57,63]. In particular, one of the main events responsible for turning on/off *MYC* transcription deals with the formation and dissipation of G4 structures. Many proteins have been identified to mediate such a dynamic regulation of *MYC* transcription by favoring the folding or unwinding of its G4 structure [57]. Among these factors nucleolin is the most extensively studied protein for its ability to promote *MYC* G4 formation [57].

Nucleolin is a multifunctional nucleolar phosphoprotein able to interact with non-conventional forms of RNA and DNA [57,63]. In particular, it has been reported that nucleolin selectively binds and stabilizes *MYC* G4, resulting in the inhibition of Sp1-induced *MYC* transcriptional activation [64]. In this context, the fluorquinolone derivative CX-3543 (quarfloxin), which has been the first-in-class G4 ligand to reach phase II clinical trials for cancer [57], has been demonstrated to affect *MYC* transcription indirectly, according to such a complex regulatory mechanism. Specifically, the drug has been reported to concentrate in the nucleolus where it binds and stabilizes a G4 structure within a ribosomal DNA (rDNA), thus impairing the interaction between nucleolin and rDNA G4. This event causes the redistribution of nucleolin within the nucleoplasm where it eventually binds to the NHE III_1_ within *MYC* P1 promoter, thus facilitating the formation and stabilization of the *MYC* G4 and, consequently, resulting in the inhibition of gene transcription [57,63,64].

Several others proteins have been found to mediated *MYC* transcription by promoting the folding (CNBP; NPM1), or by binding and stabilizing (ADAR1, LARK, mutant p53) as well as by unwinding (RecQ helicase, PARP-1, NM23-H2) *MYC* G4 [57,63]. The member of the non-metastasis 23 family of proteins NM23-H2 is able to bind the NHE III_1_ and promote *MYC* transcription by favoring *MYC* G4 unfolding [57,64]. In this context, the isaindigotone derivative compound 37 has been reported to bind with high affinity to NM23-H2 and disrupt its interaction with *MYC* G4, thus markedly inhibiting the transcription of the gene [65].

Distamycin A is a naturally occurring polypyrrole belonging to a group of antitumor antibiotics known as shape-selective molecules. They bind to the minor groove of duplex DNA at AT-rich regions leading to the inhibition of enzymes acting on DNA topology, such as helicases and topoisomerases [44]. Distamycin A has been reported to bind to the groove of G4 DNA [46,66], although there is no a general consensus on how the drug can interact with G4 DNA and several binding models has been proposed [46,67]. However, the evidence that Distamycin A can interfere with the binding of G4-associated proteins harboring specific conserved peptide motifs (e.g., nulceolin) has highlighted the potential of the drug, or its derivatives, as useful scaffold in the design of therapeutic agents targeted against specific protein-DNA G4 interactions [66]. In this context, by conjugating the anthraquinone pharmacophore with oligopyrroles, a series of “hybrid” molecules have been created with G4 dual recognition capabilities: the stacking interaction of the anthraquinone moiety to the G-tetrad and the interaction with grooves and loops of the flexible distamycin side chains [68]. In particular, 2,6-disubstituted amidoanthracene-9,10-dione based dimeric distamycin analogues were found to selectively stabilize the G4 formed within the promoter of *MYC* and inhibited DNA synthesis in a Taq polymerase stop assay performed in the presence of a 77-mer MYC quadruplex forming template and of K^+^ [68]. Although no data on the expression levels of endogenous MYC have been reported in living cells, three derivatives (ANMP, ANDP and ANTP) showed promising antiproliferative activity in selected cancer cell lines but not in normal human dermal fibroblasts [68].

Recently, a lncRNA has been also implicated in the G4-dependent regulation of *MYC* transcription. Specifically, nucleolin has been identified as the protein binding partner of LUCAT1, a lncRNA recently reported to be upregulated and to play an essential role in multiple cancer types, especially colorectal cancer [69]. In particular, it has been demonstrated that nucleolin directly binds to LUCAT1 via its putative quadruplex-forming regions and that such interaction interferes with nucleolin-mediated inhibition of *MYC* transcription [69]. Depletion of LUCAT1 results in the inhibition of colorectal cancer cells proliferation and reduced MYC expression levels, thus suggesting that LUCAT1 plays a critical role in the control of *MYC* transcripton in colorectal cancer likely by a G4-mediated inhibition of nucleolin function [69].

### 2.3. G4 Ligands with “Promiscuous” Binding Activity and/or Multiple Mechanism of Action

A variety of G4 ligands have been reported to recognize and stabilize multiple G4 targets. Such a promiscuous binding modality is likely an intrinsic feature of all G4 ligands described thus far [32], as pointed out by the evidence that the vast majority of ligands reported to interact with gene promoter G4s were primarily considered as genuine telomeric G4 ligands [14]. Though promiscuous binding may be perceived as a detrimental feature for small molecules expected to act as targeted agents, it may be regarded as a therapeutic advantage instead [14].

For instance, it was formerly reported that the tetra-substituted naphthalene-diimide derivative MM41 strongly binds the G4 within the promoters of both *BCL-2* and *KRAS* [17,70] and exerts a remarkable anti-tumor activity, with some evidence of no tumor re-growth observed after >200 days post-treatment, in a pancreatic cancer xenograft models [70]. Recently, upon a screening of imidazole-based compound library, the biimidazole derivative BIM-2 was found to selectively bind both *MYC* and *BCL-2* G4, likely through an end-stacking mode [71]. The exposure of acute myeloid leukemia (AML) cells to BIM-2 resulted in a remarkable antitumor activity as a consequence of the drug-mediated down-regulation of both MYC and BCL2, two oncogenes the over-expression of which is associated with the development of AML [71]. Similarly, an imidazole-based tanshinone IIA derivative was recently found to be able to stabilize multiple G4 targets, such as those of *MYC*, *KRAS*, *VEGF* and *BCL2*, thus resulting in the inhibition of their expression and in the arrest of triple-negative breast cancer cell growth [72].

Notably, the use of promiscuous binders may represent a clinical advantage in tumors showing acquired resistance to anticancer therapies. For instance, it was reported that a naphthalene diimide derivative was able to inhibit the growth of gastrointestinal stromal tumor cells (GIST) as a consequence of its ability to interact with the G4 located both at the telomeric level and in the promoter region of the *KIT* [17], an oncogene that is constitutively activated in GIST and it is responsible for the acquired resistance to clinically relevant tyrosine kinase inhibitors (e.g., Imatinib) [73]. Similarly, it has been reported that a NDI derivative synergistically interacted with Enzalutamide, an inhibitor of the androgen receptor (AR) signaling used in first-line therapies for metastatic castration-resistant prostate cancer, as a consequence of its ability to stabilize the G4 within the *AR* gene promoter and to remarkable reduced AR protein amounts as well as to significantly affect the expression levels of genes involved in the activation of AR program via feedback mechanisms [74]. Recently, a prolinamide-derived peptidomimetic that specifically binds to the G4 within *MYC* and *BCL-2* promoters has been reported to exert an antiproliferative activity in breast cancer cells overexpressing both genes, in comparison to cells that overexpress either of the two as well as to ligands belonging to the same family and showing a potent and specific inhibitory effect on either *MYC* or *BCL-2* transcription [75].

Besides the promiscuous binding, G4 ligands showing multiple mechanisms of action have been also described. In Section 2.1 we already mentioned the anthracyclines, a class of chemotherapeutic drugs characterized by a complex mechanism of action, which includes also G4 binding properties [45,46,47,48,49,50]. By analogy, isoindoloquinoxalin derivatives have been reported to be effective multitargeting agents showing a potent antiproliferative activity against a panel of human cancer cell lines, owing to their capability to concomitantly impair tubulin polymerization and topoisomerase I functions as well as to induce telomere dysfunctions due to their telomeric G4 stabilizing properties [76]. Similarly, a pleiotropic anticancer activity in vitro consequent to topoisomoerase I inhibition and the concomitant down regulation of MYC expression levels have been reported for a series of indenoisoquinolines, which are topoisomerase I inhibitors able to strongly bind and stabilize *MYC* G4 [77]. A type of “synergistic” mechanism of action has been recently reported also for a PARP-1 inhibitor, derived from the 7-azaindole-1-carboxamide. Other than showing PARP-1 inhibitory activity, the compound has been reported to bind and stabilize both telomeric and *MYC* G4 [78]. In this frame, it has been documented that PARP enzyme may be activated upon treatment with G4 ligand indicating the existence of an interplay between PARP-1 recruitment and G4 stabilization [78,79].

### 2.4. Targeting G4 for Synthetic Lethality

The concept of synthetic lethality refers to a genetic setting where the simultaneous occurrence of abnormalities (e.g., mutation, overexpression, or inhibition of gene function) in the expression of two or more separate genes leads to cell death; whereas abnormality in only one of the genes does not affect cell viability [80]. Since tumor cells are the result of altered gene expression, inhibitors that target the synthetic lethal partners of these mutated or overexpressed genes can lead to cancer cell death without affecting the survival of normal cells [80]. As a consequence, the synthetic lethality has a tremendous therapeutic potential in cancer [81]. Indeed, the product of a gene that has a synthetic lethal interaction with a cancer-specific somatic or germline mutation would represent a suitable candidate for drug targeting and a therapeutic agent that exploits such a synthetic lethal interaction would result in a favorable therapeutic index [81]. This notion has been elegantly exemplified by the success of PARP inhibitors in *BRCA*-mutant cancers, that has represented the first example of a synthetic lethality-based therapeutic approach, resulting in the approval of the PARP-1 inhibitor olaparib for the treatment of advanced-stage, *BRCA1/2*-mutant ovarian cancers in 2014 [80].

In this regards, G4 structures represent potential partners for synthetic lethality [82]. In particular, a synergistic interaction has been observed following the co-treatment of fibrosarcoma cells with a pyridostatin-derived G4 ligand (PDSI) and the Non-Homologous End Joining DNA repair inhibitor NU7441 [82]. In addition, PDSI exerted a greater cytotoxic effect in *BRCA2*-mutant than in *BRCA2*-proficient colon carcinoma cells [82]. This evidence suggests that PDSI-induced DNA damage is exacerbated in the presence of a pharmacological inhibitor of DNA repair, consistently with a chemically induced synthetic lethality, as well as in the context of genetically impaired DNA repair (Figure 3) [82]. Similarly, PDS has been reported to exert a remarkable cytotoxic effect in mouse, hamster and human cells lacking BRCA1, BRCA2, or RAD51 [83]. Such a toxicity extended to BRCA1-deficient cells characterized by acquired resistance to clinically relevant PARP inhibitors (e.g., olaparib) due to the depletion of 53BP1 or REV7 [83]. These findings are in trend with former evidence showing that the pharmacological inhibition of Werner syndrome helicase sensitizes cervical cancer and osteosarcoma cells to the telomeric G4 ligand telomestatin [84] as well as that the treatment of mice bearing human colon cancer xenografts with a combination of RHPS4 and a PARP inhibitor resulted in a greater reduction in tumor growth and in a longer survival rate with respect to animal that had received the single agents [79]. In addition, it has been reported that the G4 ligand CX-5461 exerts a marked cytoxic effect in *BRCA*-deficient cancer cells and in patient-derived xenograft models, including tumors resistant to PARP inhibition [85].

Although the identification of clinically relevant synthetic lethal interactions is still a major hurdle in Oncology [80,81], a genome-wide study has been recently carried out to systematically identify human genes the silencing of which promote cancer cell death in the presence of G4 ligands [86]. Genetic vulnerabilities, both in terms of genes and pathways, were indeed revealed and four genes (*BRCA1*, *TOP1*, *DDX42* and *GAR1*) were validated by an independent RNAi-mediated approach as key “G4 sensitizer” genes in melanoma and fibrosarcoma cell lines exposed to PDS or Phen DC3 [86]. Moreover, it has been reported that appropriate drug combinations can act as a surrogate for gene deficiencies in the presence of G4 ligands [86]. In this context, the pharmacological inhibition of WEE1 kinase or of deubiquitinase USP1—two newly identified G4 sensitizers [86] ‒ by MK1775 and pimozide, respectively, leads to cancer cell death potentiation when combined with the G4 ligand pyridostatin [86].

Overall, these observations have highlighted that G4 ligands may induce a synthetic lethal phenotype in cells with genetically or pharmacologically impaired pathways (Figure 3), especially DNA repair, and underscore the potential of these molecules as anticancer agents when used in rationally designed combination treatments.

## 3. Adaptive Responses in Cancer Cells Exposed to G4 Ligands

Intrinsinc and acquired drug resistance, which are based on highly complex and variable biological mechanisms, are the major causes for the failure of anticancer therapies [87]. Similarly to other anticancer agents, the therapeutic efficacy of G4 ligands may be hampered by the occurrence of drug resistance [14]. For instance, G4 ligands such as triazine and pyridodicarboxamide derivatives have been reported to be subjected to phenomena of multi-drug resistance, being recognized by efflux pumps [14]. In addition, lung cancer cells showing a resistant phenotype have been obtained upon exposure to progressively increasing amounts of a triazine-based telomeric G4 ligand [88,89]. These cells showed to be resistant to long-term ligand-mediated telomere shortening and induction of replicative senescence [88]. Notably, they showed to be cross-resistant to other telomeric G4 ligands, but not to conventional anticancer agents (e.g., doxorubicin, etoposide and Topoisomerase I inhibitors) [88]. Conversely, lung cancer cells selected upon short-term exposure to high concentrations of the triazine derivative showed cross-resistance to compounds of the same chemical family and to mitomycin C but not to other G4 ligands, indicating that such a resistance phenotype was likely restricted to triazine analogs and to DNA-damaging agents [89]. Notably, this resistant phenotype was associated with increased levels of TERT as well as altered telomere capping [89]. In addition, it has been reported that Bcl-2 overexpression was a determinant of the resistance of lung cancer cells to the triazine-mediated short-term effects, although not sufficient to confers resistance to long-term senescence induced by the same compound [90]. Moreover, it has been reported that the ectopic expression of POT1 may contribute to the resistance of human fibrosarcoma cells to telomeric G4 ligand telomestatin [91].

Overall, these findings suggest that telomere integrity, the expression of telomerase components as well as an unbalance in the expression levels of apoptotic factors may act as determinants of the resistance of cancer cells to telomeric G4 ligands [88,89,90,91].

It has been reported that cell defense response pathways (e.g., autophagy induction; the acquisition of mesenchymal traits) may become activated by cancer cells in their attempt to counteract G4-ligand-induced stress. In this regards, it has been demonstrated that melanoma cells exposed to a telomeric G4 ligand derived from the anthracene were characterized by biochemical and morphological features typically associated with autophagy, a cellular process by which cells mitigate metabolic and therapeutic stresses [92]. In particular, G4-ligand-induced autophagy was reported to occur as a consequence of DNA damage induction due to telomere uncapping [92]. Notably, the pharmacological or RNAi-mediated inhibition of autophagy resulted in a remarkable enhancement of G4 ligand cytotoxic activity, thus suggesting that autophagy may act as a safeguard mechanism to counteract telomeric G4 ligand-mediated cellular stress in melanoma cells [92].

The occurrence of the epithelial-to-mesenchymal transition (i.e., the acquisition of mesenchymal traits [87]) as a possible protective response have been observed in prostate cancer cells upon treatment with a NDI derivative able to cause the structural transition towards a G4 conformation of epidermal growth factor receptor (EGFR) gene promoter and the consequent marked reduction in EGFR protein amounts [93]. Similarly, cell plasticity, also known as phenotype switching, has been reported to represent an adaptive process by which melanoma cells adapt to treatment-induced insults [94]. In this context, it has been reported that the exposure of a mutant *NRAS* melanoma cells to a NDI derivative able to stabilize the G4 within the promoter regions of *KIT* and *BCL-2* resulted in a peculiar gene expression pattern with the over-representation of pathways, such as platelet degranulation, senescence-associated secretory phenotypes, and oxidative stress-induced senescence, that altogether indicate the occurrence of cellular/molecular changes consistent with a phenotype switching [94].

## 4. Conclusions

In the present review we have provided, without demanding completeness, a brief overlook on the potential of G4 ligands as therapeutic agents in cancer (Table 1). In particular, we draw the readers’ attention to few examples that have been paradigmatic to depict how small molecule-mediated stabilization of G4 structures may represent an intriguing strategy that could be implemented in the fight of cancer [14,15,16,17,18,29].

The variety of small molecules evaluated for their G4 stabilizing properties is countless [34]. Most of them have been reported to exert a remarkable antiproliferative activity accompanied, in some cases, by evidence of pharmacodynamic activity, when used as single agents in several in vitro models of human cancers [2,12,14,15,16,17,18,29,57,63]. Moreover, synergistic pharmacological interactions have been also documented for some G4 ligands when combined to conventional anticancer therapies [12,14], both in vitro and in vivo. In addition, the attempts made to identify G4 ligands within the available armamentarium of clinically relevant anticancer drugs have produced interesting data on the G4 binding properties of selected, though, a very few chemotherapeutic agents [47,48,49,50,56].

Despite the advancements in the field, a number of challenges still need to be fully addressed for the design of more pharmacologically relevant G4 ligands to be used as targeted therapeutic agents in cancer [32]. In this context, the primary challenges include the identification of G4 ligands with highest capability to recognize quadruplex over duplex DNA conformations and to distinguish between different quadruplex types (e.g., telomeric vs. promoter G4) or topologies [32]. In this regards, the preferential recognition of either intramolecular or intermolecular G4 has been reported to be important in determining the biological effects exerted by some ligands [14].

The identification of small molecules acting as G4 ligands requires a complex experimental workflow. The screening of drug repositories represents a feasible approach to identify lead compounds. In particular, these screenings coupled to high-throughput G4 fluorescent intercalator displacement (HT-G4-FID) method [95] have been recently documented to be a powerful tool to identify new G4 ligand scaffolds [96]. Furthermore, computational methods (e.g., molecular modeling) and virtual screening workflows have been implemented to conveniently achieve the rational design of lead compounds/scaffolds [97,98]. Whether designed with computational approaches or not, putative G4 ligands need to be subsequently assessed for their G4-binding interactions. In particular, structural analysis of the selected G4 target and of the ligand-G4 binding interactions is a prerequisite for the identification or the design of G4 binding small molecules [30]. Several biophysical methods have been implemented to this purposes, which include NMR spectroscopy, X-ray crystallography, UV/Visible spectroscopy, circular dichorisms (CD), fluorescence resonance energy transfer (FRET), surface plasmon resonance (SPR), calorimetric titration, mass spectrometry, etc., as duly reviewed in [30,31]. The combination of in silico methods with the biophysical tests have been reported to be useful for the rational design of G4 ligands [99].

Form the point of view of medicinal chemistry, the introduction of distinct chemical functionalities onto the lead/scaffold compound may be required to improve G4 ligand selectivity and/or specificity. Nonetheless, “excessive” chemical modification introduced onto the lead compound may go to the detriment of the drug-like properties of the ligand itself [30]. In fact, few G4 ligands with suitable phamacokinetic properties have been identified so far [30,31], as underscored by both the evidence that biophysical and biological/cellular data do not often correlate as expected [31] and by the irrelevant number of G4 ligands that has succeeded in entering clinical trials as anticancer agents. In this regard, the only exceptions are the fluoroquinolone derivative CX-3543 (quarfloxin), the first G-quadruplex interactive agent to enter human clinical trials [100], and its chemically related compound CX-5461 [101,102]. Both have been reported to bind and stabilize G4 DNA structures [85] and have entered phase I/II clinical trials in patients with solid or hematological malignancies (Figure 4). In particular, on the basis of the pre-clinical evidence that CX-5461 triggered synthetic lethality in tumor cells deficient for BRCA and resistant to PARP inhibitors [85], the compound has entered phase I clinical trials in 2016 for patients with solid tumors characterized by BRCA1/2 aberrations [85].

Altogether these observations clearly highlight that the identification and validation of G4 ligands as cancer therapeutic agents is not a trivial activity [5,32]. A multidisciplinary approach is highly recommended, where basic, physical and medicinal chemistry talk together fruitfully with molecular and cellular biology. The implementation of prediction algorithms [103], enabling to predict the potential formation of G4 directly from nucleic acid sequences, and the progress achieved with chemico-physical methods have provided compelling evidence on the G4 folding, structure diversity and mode of interaction with small molecules under nearly physiological conditions in test-tubes. Similarly, chromatin immunoprecipitation coupled to next-generation sequencing and the development of dedicated tools (e.g., antibodies; fluorescent probes, aptamers) for cellular imaging [25], together with the ever growing repertoire of identified G4-interacting proteins [25,33], have undoubtedly represented an important step towards a better and more accurate characterization of G4 structures and their possible interaction with ligands in the biological environment. 

However, it should be taken into account that cell factors (e.g., specific G4 interacting proteins) or cell conditions (e.g., chromatin status or transcriptional activity) may impinge on the ligand-G4 interaction [3] as well as that cell genetic backgrounds may steer the biological effects expected to arise from ligand-mediated G4 targeting in a cell type-dependent manner [14]. In this context, a comparative evaluation of the G4-ligand-mediated biological responses and observed phenotypes as a function of the different cancer cell models is of pivotal importance [14,93,94]. By analogy, the concern of safety is an issue that still needs to be better addressed. Notionally, it cannot be excluded that molecular differences (e.g., promoter epigenetic modifications, cell proliferation-dependent transcriptional activity, presence of single nucleotide polymorphysms; telomere protein composition) in normal vs. cancer cells could account for a good therapeutic index of G4 ligands. However, except for a few examples [72,74], to the best of our knowledge, a comparative evaluation of G4 ligand biological activity in tumors vs. matched normal cells have been mainly neglected. Nevertheless, evidence showing that G4 ligands impaired the growth of cancer cells without affecting the viability of normal fibroblasts or of normal cells of unrelated histology and that the antitumor activity showed by some of these compounds in in vivo models was not associated with general toxicity (e.g., body weight loss) has thus far served as an example of G4 ligand safety [14]. Furthermore, targeted delivery for the selective accumulation of G4 ligands in cancer cells has been also explored to improve their therapeutic efficacy and, hopefully, reduce possible side effects [104,105]. In particular, it has been reported that naphthalene diimide conjugated to carbohydrates (carb-NDIs, [17]) may be taken-up by cancer cells through glucose transporters, which are over-expressed by cancer cells owing to their high energy demand, and that this kind of cellular uptake correlates to some extent with the cytotoxic activity of carb-NDI [104]. In addition, the non-covalent conjugation of acridine orange-based G4 ligands with derivatives of the aptamers AS1411 has been demonstrated to be a suitable approach to achieve the selective accumulation and to improve the activity of G4 ligands in cervical cancer cells compared to non-malignant cells [105].

However, the careful consideration of the experimental settings in living systems as well as their unification and standardization is still an important requirement to enable comparison of data (Table 1) and draw proper conclusions on the antitumor activity of G4 ligands. In addition, more sophisticated models (e.g., three-dimensional and organotypic cultures; patient-derived xenografts), that address the problems of tumor heterogeneity and individual genetic backgrounds and/or of the influence from tumor microenvironment, together with the identification of reliable markers of G4 ligand pharmacodynamic activity will be useful to obtain a more realistic proof of the therapeutic potential of small molecules expected to target G4 structures in cancer.

In conclusion, targeting G4 undoubtedly represents an innovative and fascinating approach in the attempt to defeat cancer. This is an exciting field worthy of deeper investigation, not only for the design and development of novel anticancer medicaments but also to improve our knowledge of cancer biology [5].

## Figures and Tables

**Figure 1 ijms-22-05947-f001:**
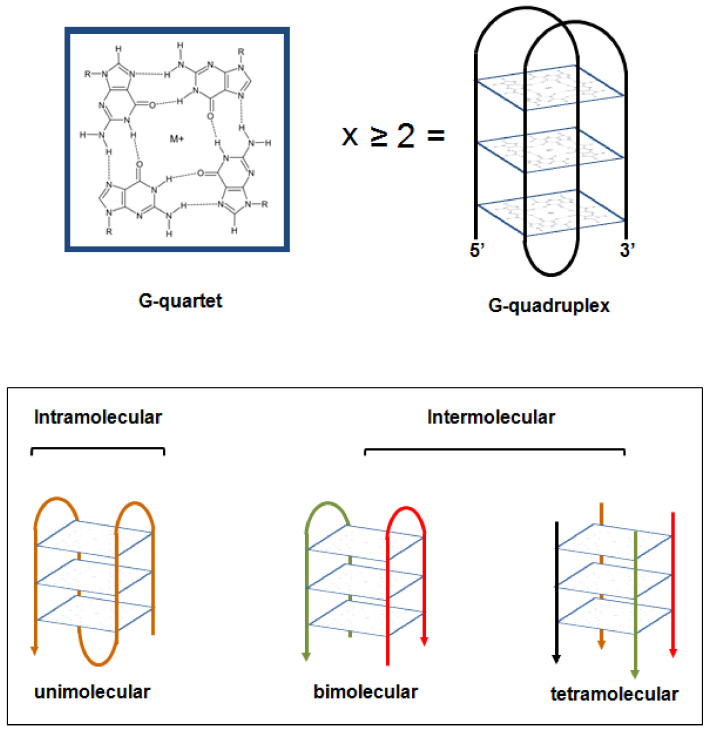
Schematic representation of a G-quartet arrangement and of a G4 structure. Examples of intramolecular and intermolecular G4 structures have been also depicted [2]. M+: monovalent cation.

**Figure 2 ijms-22-05947-f002:**
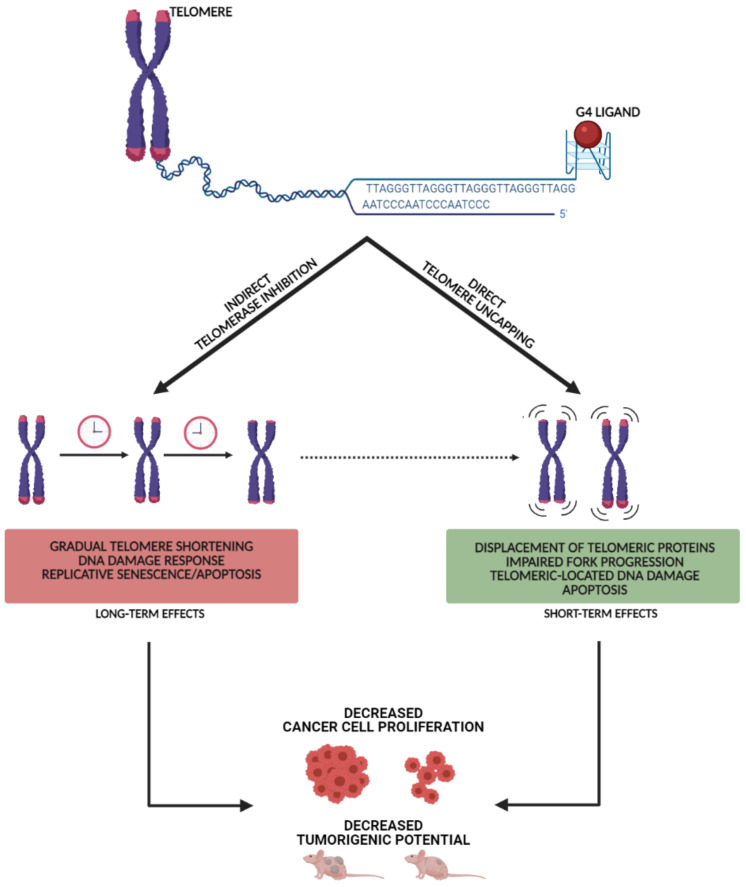
Schematic representation of the dual, sometimes interconnected (dashed arrow), path elicited by ligand-mediated stabilization of telomeric G4 (created with BioRender.com).

**Figure 3 ijms-22-05947-f003:**
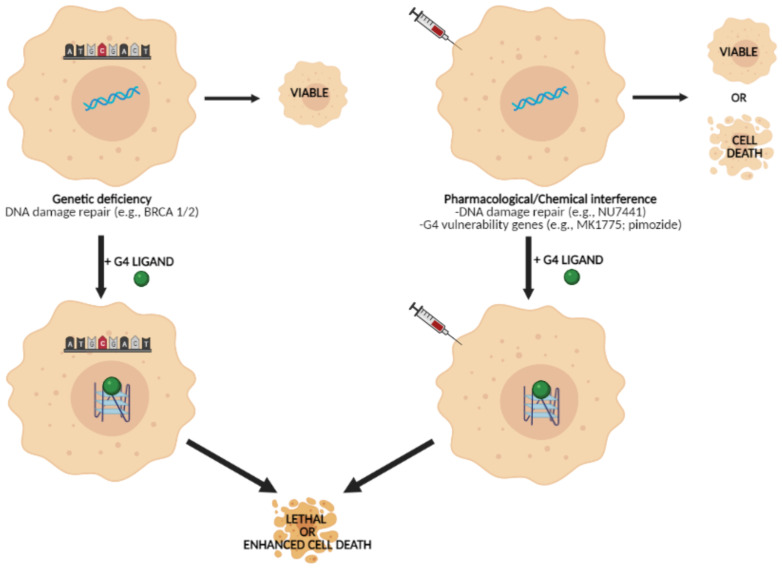
Schematic representation of G4 ligand-mediated synthetic lethality. In this context, the complementary treatment with a G4 ligand can be exploited to enhance killing of cancer cells characterized by specific genetic deficiency or upon pharmacological/chemical inhibition of the activity/expression of specific genes (created with BioRender.com).

**Figure 4 ijms-22-05947-f004:**
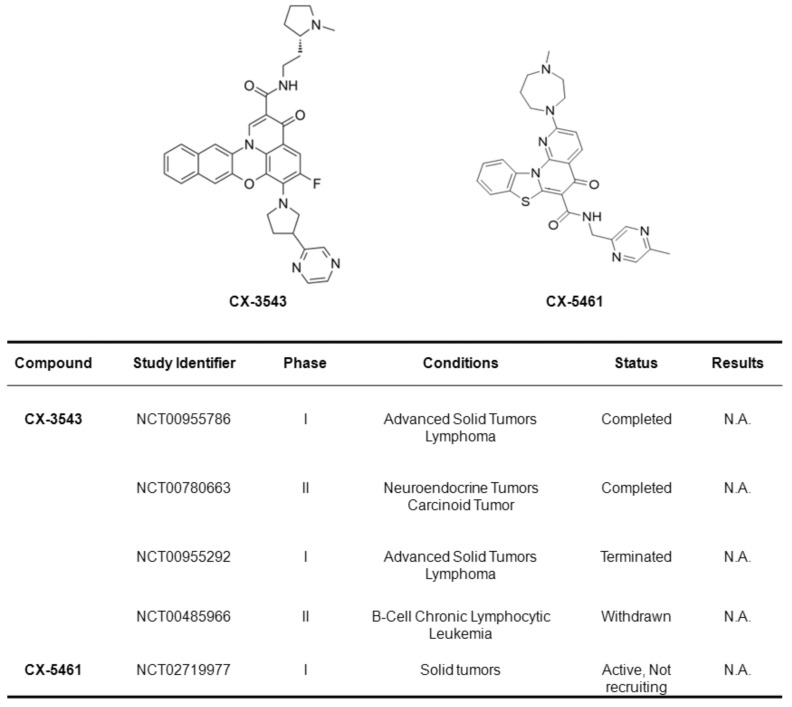
The chemical structures of the two G4 ligands under clinical testing and the related clinical trials have been reported. Study identifiers and information have been retrieved from www.clinicaltrials.gov accessed on 15 April 2021. N.A.: Not available.

**Table 1 ijms-22-05947-t001:** Summary of the biological effects of G4 ligands from selected studies discussed in the main text.

Ligand	G4 Target		Tumor Models	Biological (Anticancer) Effects	Ref
Cisplatinum derivatives (Tetra-Pt (bpy))	Telomere	G4 stabilization assessed by FRET (ΔT_m_ = 24.8 °C); binding affinity determined by SPR (K_D_ = 1.13 × 10^−7^ M); evidence by CD of parallel G4 folding under near-physiological conditions.	Osteosarcoma	-Remarkable cytotoxic activity in vitro (IC_50_ ~ 15 μM); -Inhibition of telomeric homologous recombination and suppression of ALT activity (reduced ALT-associated promyelocytic leukemia bodies; reduced c-circle DNA; reduced telomere sister chromatin exchanges) in vitro (concentration used 2 μM); accumulation of critically short telomeres after multiple population doublings; -Induction of apoptosis/senescence; -No adverse effect on normal MRC5 fibroblasts in vitro (IC_50_ = 89.3μM); -Inhibition of xenograft tumor growth in mice (20 mg/kg; i.v.); inhibition of liver metastases.	[56]
Isoindoloquinoxalin derivatives	Telomere	G4 stabilizing properties investigated by CD and FRET.	Osteosarcoma	-Cytotoxic activity in vitro (IC_50_ = 20–30 nM); -Changes in cell cycle phase distribution; induction of apoptosis; -Induction of telomere dysfunctions; -Inhibition of tubulin polymerization.	[76]
Antracene derivative (Ant1,5)	Telomere	Analyses on targeted G4 carried out in a previous study.	Melanoma	-Impairment of in vitro cell growth (IC_50_ = 4–10 μM); -Induction of telomere dysfunctions and of DNA damage; accumulation of p21^waf1^; -Occurrence of autophagy as a defence mechanism; -Increased cytotoxic activity upon pharmacological inhibition of autophagy.	[92]
Bisacridine derivatives (a9)	MYC	G4 stabilization assessed by FRET (ΔT_m_ = 9.9 °C); binding affinity evaluated by SPR (K_D_ = 7.7 μM); interaction with the G4 evaluated by CD, ITC, NMR; native PAGE and molecular docking.	Squamous cell carcinoma	-Down-regulation of MYC expression (Dual-luciferase reporter assay; RT-PCR; Western blotting); -Strong inhibition of in vitro cell growth (IC_50_ = 1.22 μM); cell cycle perturbations and induction of apoptosis; -Reduction in tumor growth in vivo (15 mg/kg; i.p.); no changes in body weight and no organ toxicity observed.	[60]
Benzofuran derivative (D089)	MYC	Analyses on targeted G4 carried out in a previous study.	Multiple myeloma	-Down-regulation of MYC expression (RT-qPCR); - Cytotoxic activity in vitro (IC_50_ = 11–50 μM); -No remarkable cytotoxic activity in HEK293T cells ectopically expressing MYC under the control of CMV promoter (IC_50_ = 50μM); -Induction of endoplasmic reticulum stress, senescence and pyroptosis in vitro.	[61]
Curcumin derivative (Cur-4)	MYC	G4 stabilization assessed by CD thermal melting (ΔT_m_ ~ 10 °C); binding affinity evaluated by steady-state fluorescence titration (K_D_ = 0.004 × 10^−6^ M) and by ITC (ΔH1 = 1.46 × 10^4^ cal/mol); increase in lifetime decay for drug-DNA complex analysed by time-correlated single photon counting; docking and molecular dynamic simulation studies.	Cervical carcinoma	-Down-regulation of c-MYC expression (qRT-PCR; Western blotting); -In vitro cytotoxic activity on cells grown as monolayer (IC_50_ = 5.0 μM); -Low cytotoxic activity in HEK293 cells (IC_50_ = 64 μM); -Decreased number of living cells in multicellular tumor spheroids with evidence of drug up-take.	[62]
2,6-disubstituted amidoanthracene-9,10-dione based dimeric distamycin analogues (ANMP, ANDP and ANTP)	MYC	G4 stabilization assessed by CD (ΔT_m_ = 3.2–11.1 °C as a function of tested compound) and Taq stop polymerization assay; binding interaction evaluated by UV-vis absorption spectral titration (Ka = 1.4–3.8 10^6^ M^−1^); fluorescence spectroscopy-based titration, ethidium bromide displacement assay, cyclic voltammetry titration; molecular docking studies.	Cervical carcinoma	-No evidence of pharmacodynamic activity in vitro (MYC expression levels were not assessed); -Cytotoxic activity in vitro (IC_50_ = 5.3–100 μM); -No cytotoxic activity on normal NIH3T3and HDFa (IC50 > 100 μM) as well as on HEK293T cells (IC50 = 15–43 μM); -Cellular morphological changes and apoptosis induction.	[68]
Indenoisoquinolines	MYC	G4 stabilization assessed by FRET (T_m_ > 5 °C in the presence of the ligand); binding interaction evaluated by NMR titration; signature of a parallel G-quadruplex assessed by CD; binding mode explored by molecular docking; binding selectivity for MYC G4 vs. KRAS G4 assessed by Competition Fluorescence Displacement.	Breast cancer	-Down-regulation of MYC expression (qRT-PCR and Western blot); -Strong topoisomerase I inhibition.	[77]
Functionalized naphthalene diimide derivatives (Compound 7)	AR	G4 stabilization assessed by FRET (ΔT_½_ = 8.3–17 °C), CD and Taq stop polymerization assay; binding affinity determined by SPR (K_D_ = 18 nM).	Metastatic, castration-resistant prostate cancer (mCRPC)	-Down-regulation of AR expression (RT-qPCR and Western blotting); -Remarkable cytotoxic acitivity in vitro; -Significant perturbations in the expression levels of KLK3 and of genes involved in the activation of AR program via feedback mechanisms; -Inhibition of telomerase activity; -Pharmacological synergistic interaction with Enzalutamide (MDV3100).	[74]
Biimidazole derivative (BIM-2)	MYC BCL-2	G4 stabilization assessed by CD (ΔT_m_ = 29.0 °C, MYC; ΔT_m_ = 18.0 °C, BCL-2); binding interaction evaluated by fluorescence titration (K_D_ = 0.75 μM, MYC; K_D_ = 1.53 μM, BCL-2); binding mode assessed by NMR titration; binding mechanism investigated by molecular modelling.	Acute myeloid leukemia	-Down-regulation of MYC and BCL2 expression (end-point RT-PCR; Western blotting); -Cytotoxic activity in vitro (IC_50_ = 9.2 μM); -No cytotoxic activity on normal BJ fibroblasts (IC50 > 40 μM); -Cell cycle perturbations with a marked increase in cells in the G0/G1 phase; apoptosis induction.	[71]
Prolinamide-derived peptidomimetic (Ligand 1)	MYC BCL-2	G4 stabilization assessed by FRET (ΔT_m_ = 15.0 °C; MYC; ΔT_m_ = 16.0 BCL–2 °C); binding affinity assessed by ITC titration (K_D_ = 1.43 μM; Δ*G* = −7.98 kcal mol^−1^, MYC; K_D_ = 2.26 μM; Δ*G* = −7.70 kcal mol^−1^, BCL-2); binding interaction assessed by molecular docking.	Breast cancer	-Down-regulation of MYC and BCL-2 expression (Dual luciferase reporter assays; qRT-PCR; Western Blotting); -Cytotoxic activity in vitro (IC_50_ = 3.8 μM); -No remarkable cytotoxic activity on normal kidney epithelial cells (IC_50_ > 50 μM); -S-phase cell-cycle arrest, DNA damage and apoptosis induction.	[75]
Core-extended naphthalene diimide derivatives	MYC BCL-2 BRAF KIT	G4 stabilization assessed by CD thermal unfolding (T_m_ > 90 °C in the presence of the ligand) FRET analysis and Taq stop polymerization assay.	Melanoma	-Down-regulation of KIT and BCL-2 protein amounts; no changes in BRAF and MYC protein levels; -Remarkable cytotoxic activity in vitro (IC_50_ = 9.0 nM and 260 nM) with evidence of G4 occurrence in cells (Immunofluorescence with a G4 specific antibody); -No remarkable cytotoxic effects on normal primary skin fibroblasts (IC_50_ > 1.000 nM); -Shutdown of RAS/RAF/MAPK and PI3K/AKT signaling pathways; -Cell cycle perturbations; induction of apoptosis (PARP-1 cleavage); Induction of a phenotypic switch in NRAS-mutant melanoma cells.	[94]
Imidazole-based tanshinone IIA derivative (Compound 4)	MYC KRAS VEGF BCL-2 Telomere	G4 stabilization assessed by FRET (ΔT_m_ = 7.89 °C, MYC; ΔT_m_ = 5.25 °C, KRAS; ΔT_m_ = 5.27, VEGF; ΔT_m_ = 4.57 °C, BCL-2; ΔT_m_ = 1.76 °C, Telomere); binding interaction evaluated by spectroscopic methods and molecular docking; Influence on G4 conformation assessed by CD.	Metastatic triple-negative Breast cancer	-Down-regulation of MYC, KRAS, VEGF, BCL-2 expression (RT-qPCR) -Cytotoxic activity in vitro (IC_50_ = 12.8 μM); -No remarkable cytotoxic activity on non-tumorigenic MCF-10A mammary epithelial cells (IC_50_ = 95.7 μM; safe index (IC_50(MCF-10A)_/IC_50(MDA-MB-231)_) = 7.48); -Cell cycle perturbations and inhibition of cell migration and invasion in vitro; -Inhibition of breast cancer growth, metastasis and angiogenesis in an in vivo zebrafish tumor model.	[72]
Tetra-substituted naphthalene-diimide derivative (MM41)	BCL-2 K-RAS	G4 stabilization assessed by FRET (ΔT_m_ = 26.4 °C, BCL-2; ΔT_m_ = 22.5 °C, k-RAS1; ΔT_m_ = 19.8 °C, k-RAS2); binding interaction evaluated by molecular modelling.	Pancreatic cancer	-Evidence of pharmacodynamic activity in vivo (reduced BCL-2 and K-RAS protein by Western blotting); -Reduction in tumor xenograft growth in vivo (10–15 mg/kg; i.v.); -No evidence of toxicity determined as absence of body weight loss; -Evidence of no tumor re-growth after > 200 days post-treatment at the dose of 15 mg/kg; -Evidence of tumor drug up-take in vivo (immunofluorescnece on tumor sections).	[70]
NDI derivative (Compound 1)	KIT Telomere	G4 stabilization assessed by FRET (ΔT_m_ = 11.2 °C, KIT-1; ΔT_m_ = 29.0 °C, KIT-2; ΔT_m_ = 28.7 °C, Telomere); binding interaction evaluated by molecular docking.	Gastrointestinal stromal tumor	-Cytotoxic activity in vitro (IC_50_ = 1.62 μM vs. 1.7 μM for Imatinib) -Nearly complete abrogation of KIT expression (RT-PCR; Western blotting) at the IC_50_ dose; -Potent telomerase activity inhibition at a sub-toxic concentration (modified/TRAP-LIG assay).	[73]
Symmetrical-and asymmetrical-substituted naphthalene diimide derivatives	EGFR Telomere	Structural transition of EGFR promoter towards a G4 conformation and stabilization of telomeric G4 evaluated by FRET, ITC and SPR titrations.	Metastatic, castration-resistant prostate cancer (mCRPC)	-Dose-dependent reduction in EGFR protein amounts; -Remarkable cytotoxic activity in vitro (IC_50_ 0.65–5.0 μM as a function of time and cell line); -Interference with RAS/RAF/MAPK and PI3K/AKT signaling pathways; -Time-dependent inhibition of prostate cancer cell growth in vitro (short-term setting); -No major changes in the rate of DU145 cell growth as well as in the amount of EGFR protein upon 60 days of weekly reiterated exposure to subtoxic amounts (½IC_50_–48h) of the ligands; -Remarkable impairment of PC-3 cell growth associated with an almost complete abrogation of EGFR protein levels upon 60 days of weekly reiterated exposure to subtoxic amounts (½IC_50_–48h) of the ligands; acquisition of mesenchymal traits and increased telomeric C-circles.	[93]

CD: circular dichroism; FRET: fluorescence resonance energy transfer; ITC: Isothermal titration calorimetry; PAGE: polyacrylamide gel electrophoresis; SPR: surface plasmon resonance.

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
