# Peer review of "On the Road to Fight Cancer: The Potential of G-Quadruplex Ligands as Novel Therapeutic Agents"

_ijms, 2021, doi:10.3390/ijms22115947_

Round 1
Reviewer 1 Report
The review 'On the Road to Fight Cancer: the Potential of G-quadruplex Ligands as Novel Therapeutic Agents' by Recagni et al. is devoted to very interested subject. Our genome consists of canonical and non-canonical elements, including G-quadruplexes(GQ) . Several G-quadruplexes are known to play a role in cancer progression, others seem to be helpful for normal cells. Can we use some GQ-ligand to fight the cancer? The answer is not obviuos. Aiming to reveal the real state in this field, I suggest to introduce more numerical characteristics. New drugs are developed to be more specific, more active and more safe. Do the GQ ligands match these criteria? I suppose the discussion of the following items would be very helpful to make the review more original compared to similar works.
Is the selectivity of the ligands sufficient to bind GQ preferably to double stranded DNA? The authors stated that the absolute specificity may be not necessary. But what selectivity is sufficient? What selectivity is achieved for known ligands?
What concentrations are necessary to see some effect of the ligands. Are they still selective to GQ at these concentrations? Are they toxic for normal cells?
How to overcome the drawbacks from low selective GQ ligands? How to target GQ in cancer cells but not in all living cells?
Please state the site of this work comparing to similar reviews in the field. For example, ref. [2] and this new review [https://molecular-cancer.biomedcentral.com/articles/10.1186/s12943-021-01328-4] seem to be overlapped by the content.
Author Response
Manuscript ID: ijms-1222508
We are grateful to the Reviewer for the positive evaluation of our manuscript. We welcome the useful tips that have been provided to improve our manuscript. The manuscript has been amended accordingly, and changes made to the text have been inserted using the Track Changes function in Microsoft Word.
Reviewer #1
The review 'On the Road to Fight Cancer: the Potential of G-quadruplex Ligands as Novel Therapeutic Agents' by Recagni et al. is devoted to very interested subject. Our genome consists of canonical and non-canonical elements, including G-quadruplexes(GQ) . Several G-quadruplexes are known to play a role in cancer progression, others seem to be helpful for normal cells. Can we use some GQ-ligand to fight the cancer? The answer is not obviuos. Aiming to reveal the real state in this field, I suggest to introduce more numerical characteristics. New drugs are developed to be more specific, more active and more safe. Do the GQ ligands match these criteria?
The discussion on the proposed items has been expanded in the conclusion section, where the evidence that the use of GQ-ligand to fight cancer is not trivial has been underscored. In addition, some numerical data from the selected papers discussed throughout the main text have been summarized in a table (see new Table 1), also to accomplish the request raised by Reviewer #2.
I suppose the discussion of the following items would be very helpful to make the review more original compared to similar works.
Is the selectivity of the ligands sufficient to bind GQ preferably to double stranded DNA? The authors stated that the absolute specificity may be not necessary. But what selectivity is sufficient? What selectivity is achieved for known ligands?
In our paper, we have used the term “selectivity” to indicate the ability of a GQ- ligand to preferentially recognized quadruplex over duplex DNA, whereas the term “specificity” was used to indicate the ability of a GQ ligand to specifically bind/recognize a given G4 target (e.g., telomere vs. MYC G4) as well as to introduce the concept of promiscuous binders. We apologize if the use we have done of these terms was confounding and incorrect and, as a consequence, to avoid confusions the terms have been deleted throughout the manuscript whenever possible and paraphrases have been used instead.
However, it should be taken into account that the terms “selectivity” and “specificity”, alongside the word “affinity”, are often used to indicate indiscriminately the preferential recognition of quadruplex over duplex DNA and the ability of a GQ ligand to recognize specific G4 targets (just to provide an example see Chem Soc. Rev 2011, 40, 5867-5892).
In our opinion, drawing proper conclusions and providing precise information on what selectivity is sufficient and achieved for the different GQ ligands described thus far in literature appear a rather difficult task to achieve for non expert in chemistry/biophysics and considering the ample number of papers published on this topic. Anyway, we have tried to address this issue in the main text, by underscoring that: i) the prerequisite for the identification of a small-molecule intended to act as GQ ligand includes the investigation of the structure of the targeted G4 and the ligand-quadruplex-binding interaction (see amended version of the conclusion section); ii) the geometry of the G4 structure is thought to allow specific recognition by small molecules by various binding modes and that a combination of two or more of these binding modes can give rise to various degrees of binding selectivity (see amended version of the introduction to section 2); iii) the presence of one or more positive charges may favor the electrostatic interaction with the negative phosphate groups of the nucleic acid backbone, though sometimes at the cost of selectivity towards specific G4 topologies (see amended version of the introduction to section 2); iv) despite the progresses in structural studies, which are important for the design of drug-like molecules specific for G4, GQ ligands with good phamacokinetic properties are still far to be clearly identified a scenario that would explain why biophysical and biological/cellular data often do not correlate as expected and and why the number of GQ ligands that has succeeded in entering clinical trials is negligible (see amended version of the conclusion section).
What concentrations are necessary to see some effect of the ligands. Are they still selective to GQ at these concentrations? Are they toxic for normal cells?
In our opinion it should be taken into account that biologically active drug concentrations required seeing some effects in cancer cells cannot be defined a priori. They indeed depend on multiple factors, including the chemical nature of the ligand, its pharmacokinetic properties (i.e., it drug-like nature), the cell experimental models used and their capabilities to mount cell defense responses as well as the occurrence of innate resistance mechanisms (e.g., multi-drug resistance). It is also straightforward that on-target effects are evaluable at sub-toxic (IC50 or lower) drug concentrations, whereas at toxic concentrations (i.e., IC80) the risk to observe off-side effects (i.e., general toxicity) increases dramatically. Concerning the therapeutic index of GQ ligands we have now highlighted that: i) it cannot be excluded that differences in cell factors/conditions could account for a lower susceptibility to GQ-ligands of normal compared to cancer cells; ii) except for a few examples, a comparative evaluation of the biological activity of GQ ligands in tumors vs. matched normal cells have been almost neglected; iii) evidence has been provided that G4 ligands may impair the growth of cancer cells without affecting the viability of normal fibroblasts or of normal cells of unrelated histology; iv) the antitumor activity showed by some of these compounds in different in vivo models was not associated with clues of general toxicity, such as body weight loss (see amended version of the conclusion section).
How to overcome the drawbacks from low selective GQ ligands? How to target GQ in cancer cells but not in all living cells?
We apologize, but we do not properly catch what the reviewer considers as drawbacks for low selective GQ ligands. However, in the original version of our paper we have outlined the concept of promiscous ligands (i.e., G4 ligands that target multiple G4s) (see amended version of paragraph 2.3)as well as of ligands with multiple mechanism of action, where the G4 binding capability is a part of a complex mechanism of action, such as the case of anthracyclines, isoindoloquinoxalin derivatives or indenoisoquinolines. In addition, we also have commented on the pivotal importance for a better understanding of the biological responses and of the observed phenotypes as a function of the different cancer cell models exposed to GQ-ligands
The specificity towards cancer cells reside on the possibilities of this approach to target sequences associated with specific cancer-associated gain-of-function (e.g., telomerase; oncogenes) as well as to target genes the protein products of which are considered non-druggable (e.g., MYC, RAS). This means that the G4 to be targeted by ligands should be selected on the basis of its/their role in cancer (i.e., oncogenes, telomere maintenance mechanisms).
In addition, besides the therapeutic index and the focus on cancer–associated targets, targeted delivery for the selective accumulation of G4 ligands in cancer (as the case of naphthalene diimide conjugated to carbohydrates or the acridine orange complexed with AS1411 aptamer derivatives) has been also explored. A discussion on this aspect has been added in the conclusion section (see amended version of the conclusion section).
Please state the site of this work comparing to similar reviews in the field. For example, ref. [2] and this new review [https://molecular-cancer.biomedcentral.com/articles/10.1186/s12943-021-01328-4] seem to be overlapped by the content.
In the present review article we provide an overview on the biological impact of targeting G4 in cancer. We have provided a brief overview of the antitumor activity exerted by some G4 ligands, by mainly focusing on telomeric G4 ligands (e.g., RHPS4) and by providing the paradigmatic example of G4-mediated regulation of MYC transcription. In this case we added also recent findings concerning the interplay between nucleolin and LUCAT1 and reported some very recent examples of novel G4 ligand designed to target MYC. Moreover, promiscuous binders and multiple targeting G4 ligands, alongside synthetic lethal interactions of G4 targeting have been also highlighted and always discussed from the biological point of view. Some evidence of the G4 binding activity of conventional anticancer agents (platinum, anthracyclines, distamycins), or their derivatives, have been also discussed. Moreover, we also devoted attention to the defense responses that cancer cell may deploy to counteract the effect of G4 ligands, an aspect that is almost completely neglected in this field. Finally we also pointed out the attention to the careful consideration of the experimental settings in living systems and on the lack of uniformity and standardization that are necessary to draw proper conclusions on the potential of G4 ligands as anticancer agents, which is still not trivial.
We are also aware that our paper does not demand completeness and that, due to the diverse aspects that characterize this specific topic and the ample number of review papers that have been published during the last few years, a certain degree of overlapping with already published articles may occur. Both papers quoted by the Reviewer have been referenced.
____
Reviewer 2 Report
This authors have reviewed G4 ligands as novel cancer therapy agents. Although the provided information is very good, this review paper still needs some sections and figures/tables to be publishable.
-The authors should add a section and talk about G4 structure. How does it happen? Any condition is required for the ligand? What types of molecules can have such a structure?
-The impact of G4 structure on therapeutic effects of different ligands should be discussed in more details. What is happening and why?
-A table including the ligands with G4 structure should be presented and discuss their properties and applications.
-How we can identify the G4 structure? How we can characterize it? A section should be added to discuss these items.
Author Response
Manuscript ID: ijms-1222508
We thank the Reviewer for the positive evaluation of our manuscript. The manuscript has been amended accordingly to the Reviewer’s suggestions, and changes made to the text have been inserted using the Track Changes function in Microsoft Word.
Reviewer #2
This authors have reviewed G4 ligands as novel cancer therapy agents. Although the provided information is very good, this review paper still needs some sections and figures/tables to be publishable.
-The authors should add a section and talk about G4 structure. How does it happen? Any condition is required for the ligand? What types of molecules can have such a structure?
In our opinion, this information was already present in the original version of the paper, and now has been a bit expanded in the revised version of the manuscript.
We need to underscore that our article was primarily conceived to provide an overview of the antitumor activity exerted by some G4 ligands, as outlined for telomeric G4 ligands and by providing the paradigmatic example of G4-mediated regulation of MYC transcription. Moreover, promiscuous binders and multiple targeting G4 ligands, alongside synthetic lethal interactions of G4 targeting have been also highlighted and always discussed from the biological point of view. Some evidence of the G4 binding activity of conventional anticancer agents (platinum, anthracyclines, distamycins), or their derivatives, have been also discussed. Moreover, we also devoted attention to the defense responses that cancer cell may deploy to counteract the effect of G4 ligands, an aspect that is almost completely neglected in this field. Finally, accordingly to the concern raised by Reviewer #1, we should also be careful to minimize overlapping with other review papers recently published on the same topic. The information required by the Reviewer may be indeed found in recent review articles that have been duly referenced.
-The impact of G4 structure on therapeutic effects of different ligands should be discussed in more details. What is happening and why?
To the best of our knowledge, there are no data on the comparative evaluation of the impact of G4 structure on the therapeutic effect of different ligands. In particular, the careful consideration of the experimental settings in living systems as well as their unification and standardization is still an important requirement to enable comparison of data and draw proper conclusions on the antitumor activity of the different G4 ligands (see new Table 1 and the amended version of the conclusion section).
-A table including the ligands with G4 structure should be presented and discuss their properties and applications.
According to the Reviewer’s request, data from the selected papers discussed in the main text have been summarized in a table (see new Table 1).
-How we can identify the G4 structure? How we can characterize it? A section should be added to discuss these items.
A detailed discussion of all these aspects in a separate section appears out of the scope of the present paper (see reply to question #1).
However, some of the details requested by the Reviewer have been outlined in the revised version of the paper (see amended version of the conclusion section). Moreover, accordingly to the concern raised by Reviewer #1, we should also be careful to minimize overlapping with other review papers recently published on the same topic.
Round 2
Reviewer 1 Report
Thank you for a careful revision of the work
Author Response
We are obliged to the Reviewer per the positive evaluation of our paper.
Reviewer 2 Report
The authors have addressed most of the concerns. The revised version of the manuscript is acceptable for publication.
Author Response
We are obliged to the Reviewer for the positive evaluation of our manuscript.